# Building a Bimodal Landscape: Bedrock Lithology and Bed Thickness Controls on the Morphology of Last Chance Canyon, New Mexico, USA

Sam Anderson[1], Nicole Gasparini[1], Joel Johnson[2]

[1]Earth and Environmental Science, Tulane University, New Orleans, 70118, USA

[2]Jackson School of Geosciences, University of Texas at Austin, Austin, 78712, USA

*Correspondence to*: Sam Anderson (sanderson@tulane.edu)

**Abstract.** We explore how rock properties and channel morphology vary with rock type in Last Chance canyon, Guadalupe mountains, New Mexico, USA. The rocks here are composed of horizontally to near-horizontally interbedded carbonate and sandstone. This study focuses on first and second order channel sections where the streams have a lower channel steepness index $(k_{sn})$ upstream and transition to a higher $k_{sn}$ downstream. We hypothesize that differences in bed thickness and rock strength influence $k_{sn}$ values, both locally by influencing bulk bedrock strength but also nonlocally through the production of coarse sediment. We collected discontinuity intensity data (the length of bedding planes and fractures per unit area), Schmidt hammer rebound measurements, and measured the largest boulder at every 12.2 meter elevation contour to test this hypothesis. Bedrock and boulder minerology was determined using a lab-based carbonate dissolution method. High resolution orthomosaics and digital surface models (DSMs) were generated from drone and ground-based photogrammetry. The orthomosaics were used to map channel sections with exposed bedrock. USGS 10 m digital elevation models (DEMs) were used to measure channel slope and hillslope relief. We find that discontinuity intensity is negatively correlated with Schmidt hammer rebound values in sandstone bedrock. Channel steepness tends to be higher where reaches are primarily incising through more thickly bedded carbonate bedrock, and lower where more thinly bedded sandstone is exposed. Bedrock properties also influence channel morphology indirectly, through coarse sediment input from adjacent hillslopes. Thickly bedded rock layers on hillslopes erode to contribute larger colluvial sediment to adjacent channels, and these reaches have higher $k_{sn}$. Larger and more competent carbonate sediment armours both the carbonate and the more erodible sandstone and reduces steepness contrasts across rock types. We interpret that in the relatively steep, high $k_{sn}$ downstream channel sections slope is primarily controlled by the coarse alluvial cover. We further posit that the upstream low $k_{sn}$ reaches have a baselevel that is fixed by the steep downstream reaches, resulting in a stable configuration where channel slopes have adjusted to lithologic differences and/or sediment armour.

## 1 Introduction

Many studies have recognized that lithologic contrasts are expressed in topography (e.g., Howard and Dolan, 1981; Duvall et al., 2004; Johnson et al., 2009; Hurst et al, 2013; Johnstone and Hilley, 2015; Harel et al., 2016). For example, Wohl et al. (1994) found that knickpoints in the Nahal Paran River, Israel formed where relatively resistant chert layers were exposed. River channels may narrow in reaches with harder rocks (e.g., Bursztyn et al., 2015; Montgomery and Gran, 2001) and/or

steepen (e.g., DiBiase et al, 2018; Darling and Whipple, 2015). The properties that control bedrock erodibility (such as intact
rock strength, fracture density, and bedding dip) influence both rates of channel adjustment and how channel and hillslope
morphologies evolve through time (e.g., Weissel and Seidl, 1997; Wolpert and Forte, 2021; Chilton and Spotila, 2022).

Erodibility is a model-dependent parameter. For example, the stream power (or shear stress) erosion model can be written

as

$$S = \left(\frac{E}{K}\right)^{\frac{1}{n}} A^{-\frac{m}{n}} \qquad (1)$$

where $K$ is fluvial erodibility, $S$ is channel slope, $E$ is erosion rate, $A$ is drainage area, and $m$ and $n$ are exponents that can be
calibrated to local conditions (e.g., Whipple and Tucker, 1999).  This model assumes that erosion rates can be approximated
by a power law function of reach slope and drainage area (e.g., Howard, 1994; Stock and Montgomery, 1999). This
approximation may be adequate to describe multiple processes (Gasparini and Brandon, 2011). The model is widely applied
in tectonic geomorphology to infer relative erosion rates, although the $E/K$ ratio shows that it is equally sensitive to erodibility
differences (e.g., Whipple and Tucker, 1999, Wobus et al. 2006). Whipple and Tucker (1999) show that $K$ is a function of not
only bedrock properties but also channel geometry, basin hydrology, and sediment load; nonetheless the dependence of $K$ on
bedrock properties arguably remains the largest unknown.

Using the simple and idealized stream power model (Equation 1), Forte et al. (2016) and Perne et al. (2017) demonstrated

that spatial contrasts in bedrock erodibility can result in complex and sometimes counterintuitive relations between local
erosion rate, channel slope, and bedrock erodibility. These include local erosion rates being higher in stronger (less erodible)
bedrock layers compared to weaker layers, channels evolving to be steeper in weaker bedrock, and a steady-state topographic
configuration being unattainable at the spatial scale of erodibility contrasts (when measuring elevations and erosion rates
vertically). Perne et al. (2017) showed that local channel topography tends to evolve towards an "erosional continuity" steady
state in which layers with contrasting erodibilities have equal erosion rates when measured parallel to lithologic contacts, but
that topographic steady state in which erodibility contrasts are expressed in landscapes is only strictly possible for vertical
contacts. Erodibility contrasts oriented perpendicular to vertical—i.e., horizontal layers— "exhibit the largest departures from
steady-state, and the most complex patterns of landscape evolution" (Forte et al., 2016).  An advantage of studying
approximately horizontally layered rocks is that the spatial pattern of erodibility contrasts is predictable. Thus, idealized models
suggest that strong erodibility contrasts from horizontal rock layers can be expressed in topography in complex but potentially
understandable ways.

A fundamental challenge in moving from models to field constraints is that many variables influence rock erodibility.

Fluvial erosion processes, including abrasion (impact wear) and hydraulic block plucking, depend on rock properties in
different ways and make the relationship between overall erodibility and measurable variables nonunique. For abrasion from
impacting grains, bedrock incision rate should scale inversely with rock tensile strength (Sklar and Dietrich, 2001; Mueller-
Hagmann et al., 2020). Fracture density influences bedrock incision rates and dominant processes, especially block plucking
(e.g., Spotila et al., 2015; Dibiase et al., 2018; Scott and Wohl, 2019 ESPL; Chilton and Spotila, 2022). It remains unclear how
to quantitatively relate different rock properties to erodibility in different settings; semiquantitative relations have been
proposed but not widely validated for fluvial settings (e.g., Selby, 1982).

Channel morphology adjusts not only to substrate erodibility, but also to transport the imposed abundance and size

distribution of sediment (e.g., Hack, 1957). Importantly, in erosional landscapes the sediment size distribution can reflect
bedrock properties, as it derives primarily from hillslope erosion in the upstream watershed (Thaler and Covington, 2016;
Shobe et al., 2021b). Mechanistically, abrasion requires sediment transport (tools effect), while incision by most erosion
processes is inhibited by alluvial cover (cover effect) (Sklar and Dietrich, 2004). Studies have found that the abundance and
size distribution of sediment delivered to a channel reach from upstream and surrounding hillslopes can steepen reaches beyond
what might be predicted from channel bedrock properties alone (e.g., Brocard and van der Beek, 2006; Johnson et al., 2009;
Thaler and Covington, 2016; Chilton and Spotila, 2020; Lai et al., 2021; Shobe et al 2021a). In particular, Thaler and Covington
(2016) isolated the role of large and relatively immobile boulders on channel slopes by comparing reaches incised into the
same underlying bedrock, but with different amounts and sizes of boulders supplied from a caprock layer present in only some
watersheds. Further, Shobe et al. (2021a) developed a steepening ratio, that calculates the impact of boulders on channel slope
in comparison with a boulder free reach. Discharge variability has also been shown to matter for understanding cover effects
in natural systems, particularly in reaches with boulders, as the bigger the boulder the larger (and more rare) the flood that can
mobilize it larger boulders are (e.g., Lague et al., 2005; Shobe et al., 2021b; Ramming and Whipple, 2022). Importantly, the
landscape evolution models used by Forte et al. (2016) and Perne and Covington (2017) did not include sediment load, and it
remains unclear how cover effects and boulder supply may influence relations between topography and bedrock properties in
natural landscapes. Taken as a whole, the studies above suggest that rock properties impact erosion processes and channel
morphology in multiple ways. Strength and resulting erosion processes are impacted by the density of fractures and the relative
dip of the bedding. Fracture density also influences size distributions of coarse sediment supplied to channel reaches.

The overall objective of this study is to better understand how fluvial network topography in a real erosional landscape is

influenced by horizontal rock units, both directly through bed erodibility and indirectly through coarse sediment supplied from
hillslopes. We hypothesize that local topography—as quantified through channel steepness index ($k_{sn}$, defined below) and local
relief—correlates with measurable properties of both bedrock and boulders. The field area has alternating layers of primarily
sandstone and primarily carbonate rocks. Our approach was to measure compressive rock strength, fracture density, boulder
dimensions, and bedrock exposure along channels from extensive field surveys. We objectively quantified rock mineralogy
from field samples. We do not have measurements of erosion rates and so cannot directly calculate erodibility (Equation 1).
However, we interpret that patterns of bedrock-controlled erodibility and boulder distributions in this landscape have resulted
in a bimodal topography. Upstream channels and hillslopes have lower channel steepness, gentler hillslopes, and hypothesized
higher erodibilities. Downstream channels and hillslopes are steeper, with hypothesized lower erodibilities.

## 2 Field Area

This study focuses on channels with intermittent flow in Last Chance canyon, which is part of the Guadalupe mountains (Figure 1). During Permian time, a shallow lagoon existed behind a reef complex to the south and deposited what would become interbedded carbonate and siliciclastic bedrock of Last Chance Canyon (Hill, 2000; Phelps et al., 2008; Kerans et al., 2017). The Guadalupe mountains were uplifted during basin and range extension beginning 27 million years ago, exposing the previously buried bedrock (Chapin and Cather, 1994; Ricketts et al., 2014, Hoffman, 2014; Decker et al., 2018).

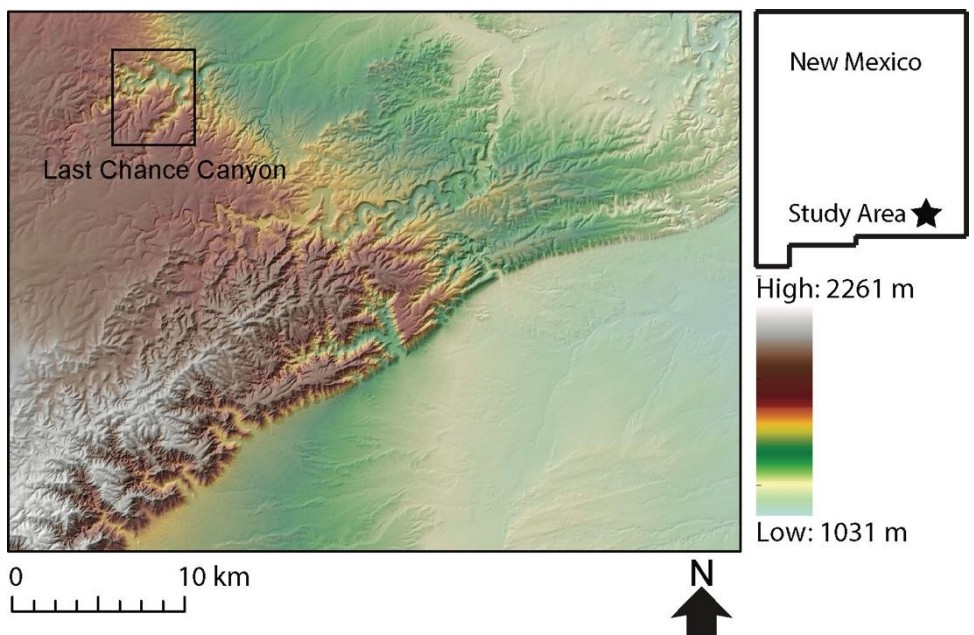

**Figure 1: Regional topographic map of a section of the Guadalupe mountain range, with location in New Mexico, USA, shown at right.**

Because of its morphology and accessibility, we collected data along tributaries of Last Chance Canyon to identify how changes in bedrock lithology and boulder characteristics correlate with stream channel and landscape morphology. Over the small spatial area and range of vertical elevations of the specific study channels (Figure 2), climate varies minimally. Mean annual precipitation is ≈40-50 cm/year and mean annual temperature ≈14-16 °C (PRISM Climate Group). Last Chance Canyon has horizontally to near-horizontally bedded bedrock and is currently tectonically inactive (Hill, 1987; Hill, 2006). Mapped descriptions of stratigraphic units in Last Chance canyon include both sandstone and carbonate bedrock, with bed thicknesses within mapped units on the order of centimetres to meters (Figure 2; Scholle et al., 1992; Hill, 2000; Phelps et al., 2008), which agrees with what we observed in the field (Figure 3). This seemingly simple variation in lithology makes Last Chance canyon an ideal location to explore the effect of varying bedrock properties on stream channel morphology.

Beyond Last Chance Canyon, the Guadalupe Mountains are comprised mostly of horizontally to near-horizontally bedded
carbonate and siliciclastic rock (Figure 2). Rock unit descriptions from published maps are not at the scale needed for us to
constrain rock strength variability along channels (NPS, 2007). Higher order channels further downstream of the survey
reaches in Last Chance Canyon are inundated with coarse alluvium and have essentially no exposed bedrock. Therefore, we
focus on first- and second- order channels, as defined by Strahler (1957), in Last Chance Canyon, because this is where we
have collected extensive data and where we are able to measure rock properties in the channel bed. Although some of our
observations from Last Chance Canyon likely apply in other locations, mapped rock units have spatial variability in rock
properties, and we refrain from making conclusions about other parts of the landscape.

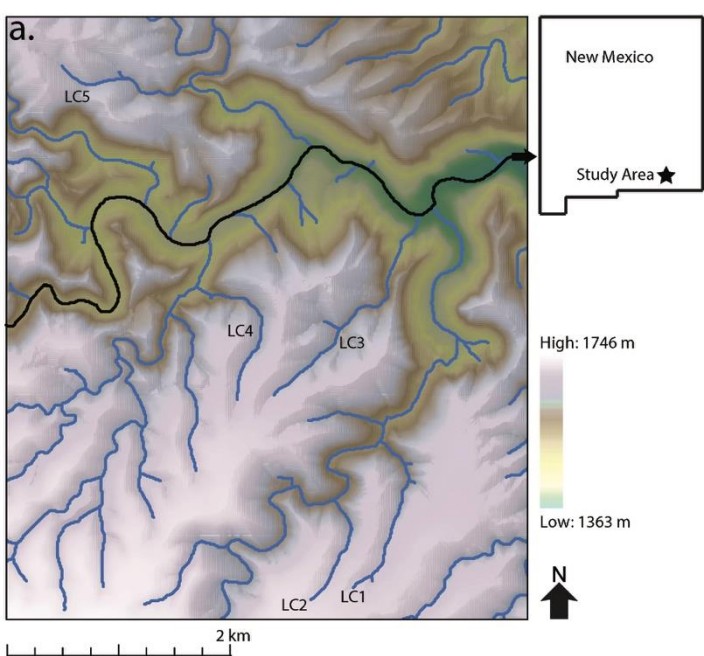

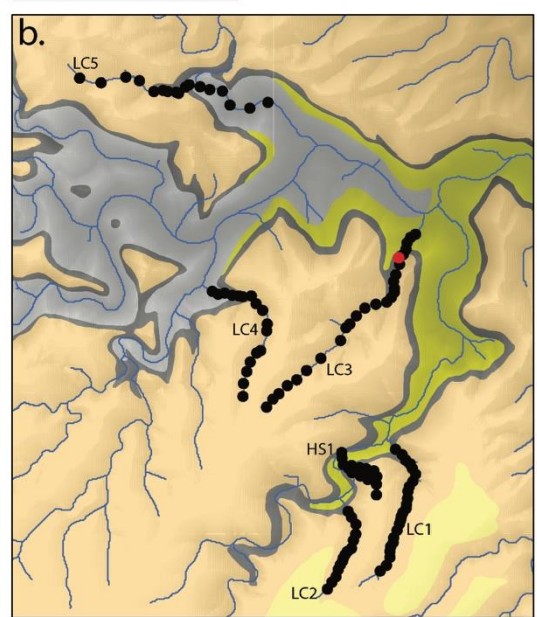

- ● Sampled Locations

### c.

| | Rock Unit | Description | Approximate Elevation (m) |
|---|---|---|---|
| | Queen Formation | Predominently sandstone with some dolomite near the base of the unit. | 1700 up |
| | Greyburg Formation | Mostly 2.5 to 15 cm thick sandstone beds with few 2.5 cm to 3 m thick dolomite beds | 1540 - 1560 |
| | Upper San Andres Formation | 0.5 cm to 1 m thick dolomite beds with one to three sections of thinly bedded sandstone. | 1440 - 1510 |
| | Lower San Andres Formation | 0.3 to 1.5 m thick dolomite beds with some medium to very grained sandstone beds. | Varies |
| | Sandstone tongue of the Cherry Canyon Formation | Very fine grained, well sorted quartz sandstone with scattered, irregular chert nodules. | 1400 |

Approximate thickness (m): 150 m / 30 m / 0 - 130 m / 0 - 150 m

**Figure 2: a. Topographic map with elevations superimposed on a hillshade of Last Chance canyon with five ephemeral study channels LC1 – LC5 labelled. Main stem channel that all streams flow to is coloured black with arrow indicating the direction of stream flow. All mapped streamlines begin with a threshold drainage area of 1 km². b. Geologic map of study area with c. a description of mapped lithologies (King, 1948; Boyd, 1958; Hayes, 1964; USGS, 2017). Approximate elevation and thicknesses apply only to the section of Last Chance canyon displayed here. Dots in b indicate locations we took measurements at (in five tributaries, labelled LC1-LC5and one hillslope labelled HS1). The reach marked with a red dot is LC3.2 and is shown in Figure 4.**

## 3 Methods

### 3.1 DEM Analysis

We used a 10 m digital elevation model (DEM) of Last Chance canyon to identify channels of interest to survey and to calculate relevant topographic metrics, and slope breaks along longitudinal stream profiles (USGS, 2019). The normalized channel steepness index, $k_{sn}$, is a measure of channel gradient normalized for drainage area (i.e., in principle allowing reach slope to be compared independent of drainage area):

$$S = k_{sn}A^{-\theta_{ref}} \qquad (2),$$

where $\theta_{ref}$ is a reference concavity (Whipple and Tucker, 1999; Wobus et al., 2006). Based on a calibration to this landscape we use $\theta_{ref} = 0.5$, giving m$^{-1}$ as the units for $k_{sn}$. Although $k_{sn}$ is an empirical metric of fluvial topography (Equation 2) and not model dependent, if the stream power model is assumed to be valid then combining Equations (1) and (2) gives $E/K = k_{sn}^{n}$, Illustrating how this topographic metric potentially informs both erosion rates and erodibilities. $k_{sn}$ allows for the comparison of slope along a single channel or among multiple channels to isolate erosional and/or bedrock erodibility patterns (Kirby & Whipple, 2012). We also calculated χ plots (Perron and Royden, 2013; Willet et al., 2014), which represent a method of transforming the horizontal variable (x) of longitudinal stream profiles into dimensionless variable χ. Generally speaking, a smoothly concave stream profile without changes in erodibility or erosion rate along its length will be a straight line on an elevation vs. χ plot, while deviations from linear may represent changes in erodibility or erosion rate (Perron and Royden, 2012; Willet et al., 2014). Because channels can adjust to more resistant lithologic units by steepening across them (Duval et al., 2004; Jansen et al., 2010), we used χ plots and $k_{sn}$ maps to detect changes in slope that could be due to differences in bedrock erodibility and/or sediment size and cover. TopoToolBox and Matlab were used to generate longitudinal profiles, $k_{sn}$ maps, and χ (chi) plots of all surveyed channels (Schwanghart and Scherler, 2014).

We also used a DEM to measure channel slope and hillslope relief. Elevations were measured 75 m upstream and 75 m downstream from each reach, the downstream elevation was then subtracted from the upstream elevation and the value was divided by the length, 150 m, to determine slope. The 150 m scale of measurement was used to smooth the data, as is commonly done in topographic analysis because slope data can be noisy and have artifacts (Wobus et al., 2006; Kirby and Whipple, 2012). Relief was measured in ArcGIS using a circular 500 m window around each reach. The radius of the relief window was chosen because ridgetop spacing is ~ 500 m in the field area. Therefore our relief values roughly represent the elevation change from valley bottom to ridge top.

## 3.2 Field Surveys

In March and May of 2018, and in February of 2021, we surveyed five channels which we had preselected based on DEM analysis, mapped geology, and accessibility. Our investigation started in lower order channels at elevations above 1400 m in channels LC3, LC4, and LC5 and in elevations above 1500 m in channels LC1 and LC2 (Figure 2). We studied reaches of varying length in the five different channels. USGS topographic contour maps of the field area use a 40 ft (≈12.2 m) contour interval. Following these maps for convenience and to ensure unbiased sampling, at every ≈12.2 m contour interval we surveyed channel reaches for bedrock properties when exposed, measured the largest, assumedly most immobile, boulder in the reach, and took rock samples from each to confirm minerology. Previous work suggests that boulders and the coarsest sediment size fractions can significantly influence reach topography, erosion, and transport (e.g. Shobe et al., 2016). The largest boulder was chosen (rather than a particular coarse grain size percentile such as D84) as a balance between available time for field surveys and statistical accuracy for characterizing coarse sediment. We assume that the largest boulder size is positively correlated with other coarse grain size percentiles when averaged over many surveyed reaches, while acknowledging that this method may introduce a bias due to size selection. For each boulder we measured the longest (a), intermediate (b) and shortest (c) axes (Figure 3). We multiply these dimensions together to approximate boulder volumes. We also constrain differences in boulder shape using a simple shape factor defined as c/a (the shortest axis divided by the longest axis)

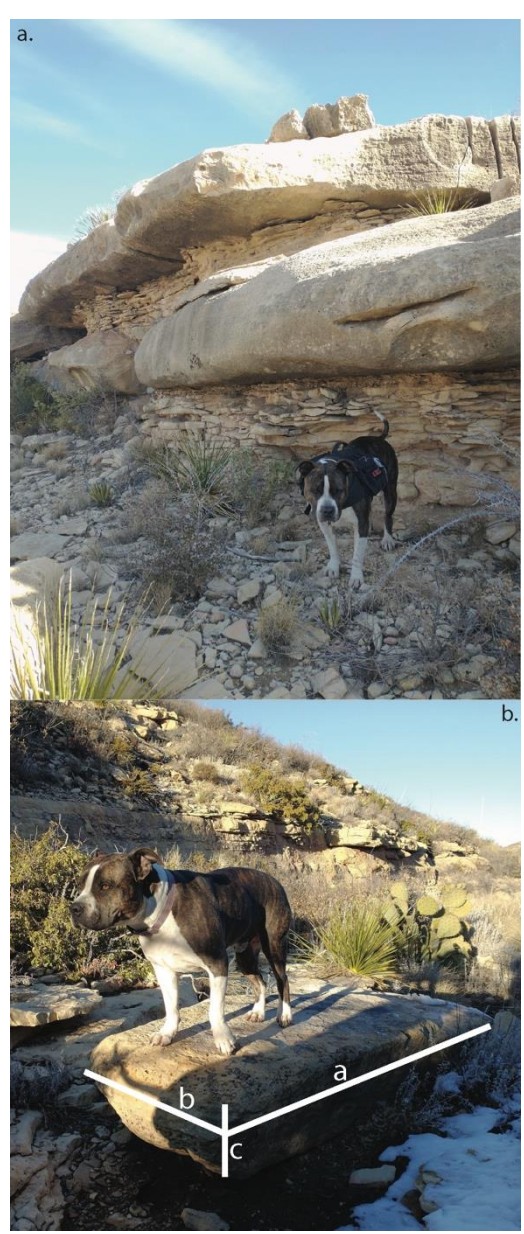

**Figure 3: Photo demonstrating the differences in a. bed thicknesses between lithologies and b. large boulders (with axes labelled in white) sourced from the more thickly bedded dolomitic rock. Dog height is approximately 75 cm at shoulders.**

## 3.3 Bedrock Properties and Photogrammetry

We used a Schmidt hammer to take a minimum of 30 rebound values in each reach we surveyed that had exposed bedrock (Niedzielski et al., 2009). Schmidt hammer rebound values scale with compressive strength but are typically reported as unitless numbers between 10 (very weak) and about 70 (very strong) (e.g., Bursztyn et al., 2015; Murphy et al., 2016). We discarded Schmidt hammer values less than 10, the minimum value the device can read, as they represent multiple values and

make statistical analysis of the data difficult (Duval et al., 2004). Schmidt hammer values were recorded at roughly evenly
spaced intervals up the thalweg of each channel regardless of weathering or presence of fractures. All Schmidt hammer values
were taken perpendicular to the bedrock surface. Schmidt hammer values are affected by proximal discontinuities. Because
we sampled at evenly spaced intervals in the exposed bedrock and did not avoid discontinuities, our Schmidt hammer values
reflect a combination/distribution of local rock elastic properties modulated by discontinuities (Katz et al., 2000). We used
two-sample, two-tailed t tests to determine to determine if rebound values differ between rock types and between the steep
downstream and shallow upstream channel sections were different or similar.
We used a GoPro5 attached to the end of a selfie stick to take wide-angle HD videos of the bottom of 18 different reaches
of varying size. We used iMovie to extract frames (1 frame for every second of video). We used Agisoft PhotoScan (Agisoft
PhotoScan Professional, 2018) to generate high resolution orthomosaics. First we aligned the frames from the GoPro videos,
then built a dense cloud, created a DSM (called a DEM in Agisoft PhotoScan), and finally made an orthomosaic.
Discontinuities were visually interpreted and manually traced on the orthomosaic images using Adobe Illustrator software
(Figure 4). Bedding planes are zones of weakness by which bedrock can be plucked, and both bedding planes and fractures
were treated as discontinuities (Spotila, 2015). Although identifying discontinuities from the images was somewhat subjective,
the same person did all these analyses and so they are likely internally consistent. We used Fraqpac (Healy, 2017), a Matlab
software suite, to determine the discontinuity intensity, which is the length of all traced discontinuities divided by the area
examined in each reach. The discontinuity intensity is reported in units of per meter.

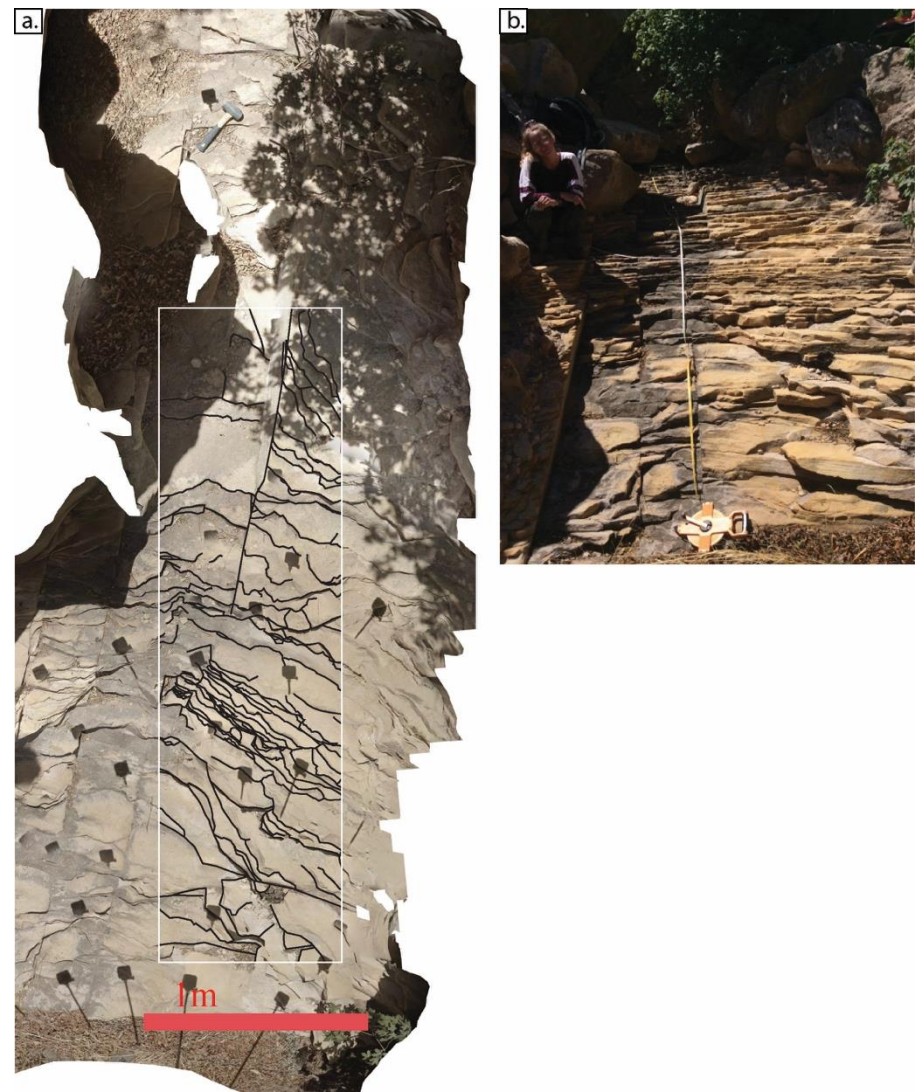




**Figure 4: a) An orthomosaic and b) photo of sandstone reach LC3.2 (Figure 2b), with a discontinuity intensity of 13.03 1/m in the steep channel section. The shadows in the orthomosaic are from the GoPro and selfie stick used to film the reach. Lat, Long: 32.252513, -104.701289**

We used a drone, DJI Mavic 2 pro, to take photos of the five surveyed channels from elevations of approximately 20 meters above the five stream channels, and 120 meters above adjacent hillslopes for three of the five channels. We used Agisoft PhotoScan to generate high resolution digital surface models (DSMs) with 0.027 to 0.28 m resolution (we refer to these as DSMs rather than DEMs because vegetation is not removed from the DSMs) and orthomosaics of the five channels and three adjacent hillslopes. The methodology we used to create the DSMs and orthomosaics is the same that we used to create the orthomosaics of the reaches and is described in the previous paragraph. We used the orthomosaics to quantify relative

proportion of where stream channel beds were exposed bedrock or covered with sediment. Given the sub-decimeter scale of our channel imagery, it was generally clear what was and was not sediment on the channel bed, and we did this mapping by eye. We partitioned the channel reach into lengths that were and were not covered in sediment. This means that we only looked at changes along the channel center line. However, this seemed a reasonable assumption as the predominant variation in sediment cover was usually down channel, not across channel.

### 3.4 Lithology

At each ≈12.2 M elevation contour interval we collected rock samples from exposed bedrock and from the largest boulder in the stream channel to ensure correct categorization of lithology. The minerology of each rock sample was assumed to be representative of the minerology of the reach or boulder it was taken from. Our efforts to determine end-member lithological classifications of sandstone or carbonate in the field were imprecise because individual samples usually contained both carbonate and quartz. To find a quantifiable ratio of the amount of carbonate in each sample, back in the lab we broke off a very small piece of each rock sample that appeared representative of its composition and ground up this subsample using a jaw crusher and disk mill. The average size of each subsample that we processed was 1.689 g with a standard deviation of 0.707 g, and the scale was precise to 0.001 g. The ground subsample was rinsed in water a minimum of five times, dried in an oven overnight, and then weighed the following morning. We then dissolved the carbonate minerals by soaking each sample in Nitric acid for at least 24 hours. The subsample was again rinsed in water a minimum of five times and dried overnight. We used a microscope to check that only quartz remained after dissolving each subsample in nitric acid. We then reweighed each subsample to determine the ratio amount of dissolved carbonate minerals. Samples were classified as carbonate if the subsample had more than 50% carbonate minerals, and sandstone if they had more than 60% quartz (Bell, 2005). Samples which ranged from 50 – 59% of quartz were lithologically unclassified, so that the endmember carbonate and sandstone classes would be more distinct. However, the fact that there was bedrock exposed was still recorded. Only 1 bedrock sample and 2 boulder samples fell in the range of 50-59% quartz, compared to 56 boulder and 56 bedrock samples that were classified. To ensure the validity of this methodology, we replicated this process on six samples by repeating the process with a different subsample from the original rock sample. For one of the samples, we replicated this process five times. All replicate measurements demonstrated similar results (standard deviation of 0.62% carbonate dissolved, and variance of 0.39% carbonate dissolved).

### 4 Results

### 4.1 Morphometric Analysis

Last Chance canyon tributaries have upstream sections with relatively shallow channels and lower gradient hillslopes, and a knickzone downstream which has steep channels and hillslopes (Figure 5). χ plots (Figure 5c and d) and field observations demonstrate that the stream channels transition from steep to shallow at approximately 1640 m for channels 1 and 2 and at

approximately 1550 m for channels 3, 4 and 5. At the transition from steep to shallow in channels 1 and 2 the slope of the $\chi$
plot changes less than in channels 3, 4, and 5. The average value for slope gradients above 1550 m in elevation is 16.5 (n =
145765, $\sigma = 11.1$), above 1640 m in elevation the average slope is 11.5 (n = 68853, $\sigma = 8.8$), and from 1400 m to  1550 m in
elevation the average slope gradient is 24.5 (n = 70438, $\sigma = 11.1$).

We used a t test to verify a bimodal distribution of hillslopes between the shallow section, elevations above 1550 m in

channels 3, 4, and 5 and above 1640 m in channels 1 and 2, and the steep section, elevations from 1400 to 1550 m. The null
hypothesis was that the hillslope values in the steep and shallow sections are the same and/or do not vary between the lower
steepness (upstream) and higher steepness (downstream) reaches. This would indicate that landscape form does not change at
the elevations we interpreted using the chi plots in figure 5. Conversely, if the hillslope values from the different elevation bins
are from statistically different populations, this supports our interpretation that landscape form changes at elevation 1550 m in
channel 3, 4, and 5 and 16 40 m in channels 1 and 2. The t test ($t = -155.4$, *t critical* $= 1.96$, $\alpha = 0.05$) demonstrated that slope
gradient values from the shallow channel section are different that slope gradient values from the steep channel section.

We do not have erosion rate data for the field channels, and so cannot quantitatively constrain erodibility (Equation 1).

Our overall approach instead is to evaluate whether the existing fluvial morphology in this part of the landscape likely reflects
measurable rock properties.

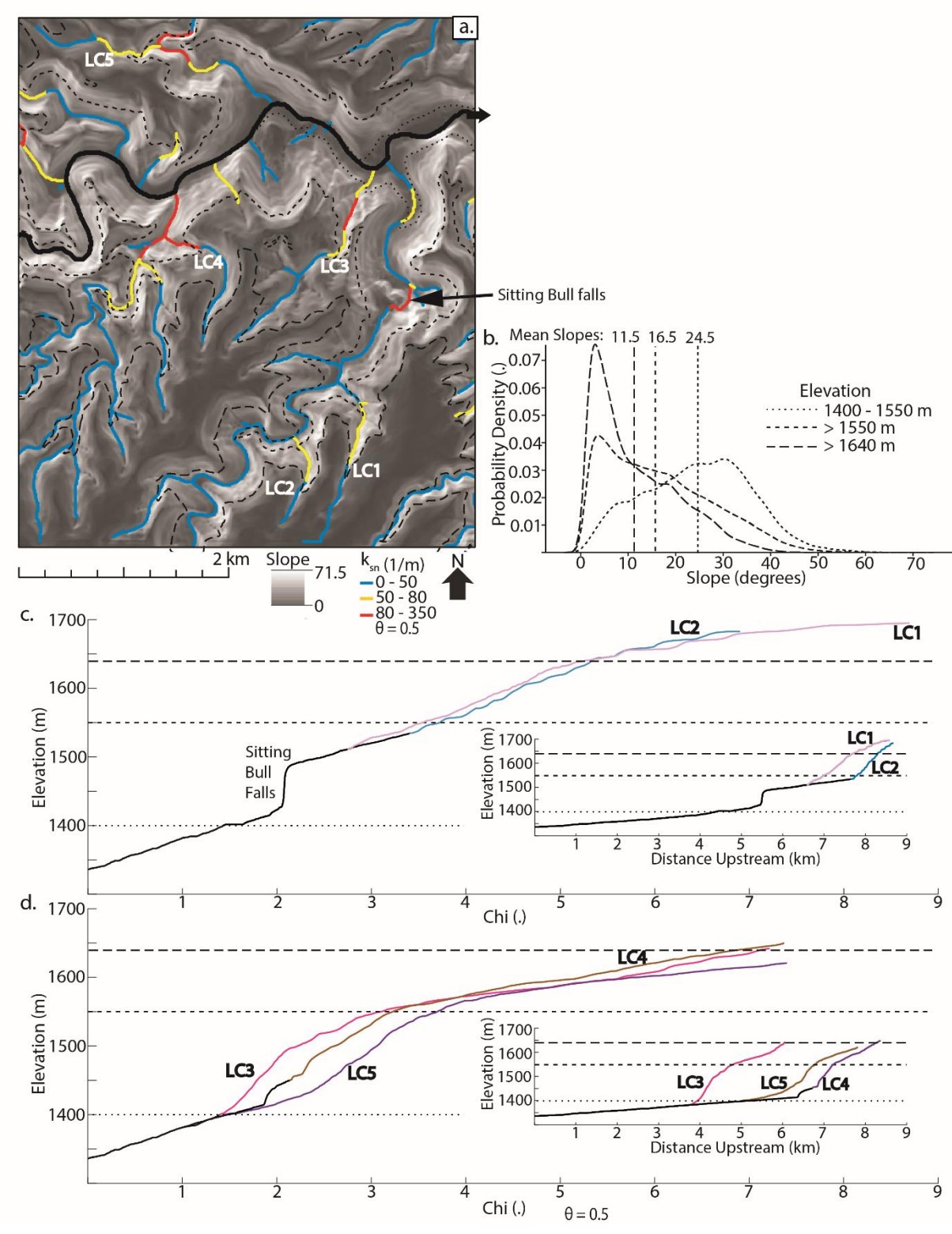

 **Figure 5 - a. Slope map of Last Chance canyon with channel colored by $k_{sn}$ values. The contour lines correspond to elevations which are interpreted as approximate inflection points for hill and channel slope (1550 m for LC 3, 4, and 5 and 1640 m for LC 1 and 2). b. Kernal density estimates of slope values from the shallow landscape sections, >1640 m and > 1550 m, and the steep section, 1400 to 1550 m. c. χ plots of LC1 and LC2 and d. LC3, LC4, and LC5 with inset of channel profiles. The downstream portion of the channels that is colored in black in c and d was not surveyed.**

### 4.2 Bedrock Properties

The extent of exposed sandstone and carbonate rock in the five study channels is presented in Table 1. The data are presented for above and below 1550 m elevation, of the elevation in which the channel steepness index changes in LC 3, 4, and 5. Due to limits on our field time, there are a reaches of exposed bedrock above 1550 m that we were not able to sample, and these are labelled as "undefined rock". In all the channels except LC1 there is more alluvial cover downstream of 1550 m than above 1550 m.

| | Above 1550 m | | | | | |
|---|---|---|---|---|---|---|
| | Exposed Carbonate | Exposed Sandstone | Exposed Undefined Rock | Alluvial cover | Mean Boulder Volume (m$^3$) | Boulder Standard Deviation (m$^3$) |
| LC1 | 1.4% | 4.4% | 0.0% | 94.2% | 1.3 | 2.2 |
| LC2 | 7.5% | 1.1% | 1.3% | 90.2% | 0.3 | 0.1 |
| LC3 | 2.8% | 10.0% | 19.9% | 67.3% | 0.2 | 0.2 |
| LC4 | 15.7% | 8.3% | 4.8% | 71.2% | 0.6 | 0.8 |
| LC5 | 13.8% | 6.9% | 17.8% | 61.5% | 0.5 | 0.7 |

| | Below 1550 m | | | | | |
|---|---|---|---|---|---|---|
| | Exposed Carbonate | Exposed Sandstone | Exposed Undefined Rock | Alluvial cover | Mean Boulder Volume (m$^3$) | Boulder Standard Deviation (m$^3$) |
| LC1 | 18.2% | 7.8% | 0.0% | 74.0% | 2.7 | 2.7 |
| LC2 | 0.0% | 0.0% | 0.0% | 100.0% | 0.4 | 0.1 |
| LC3 | 14.0% | 0.8% | 0.0% | 85.2% | 4.4 | 3.8 |
| LC4 | 8.0% | 0.0% | 0.0% | 92.0% | 11.9 | 12.7 |
| LC5 | 18.6% | 2.2% | 0.0% | 79.2% | 15.8 | 21.5 |

**Table 1 – Table describing channel lithology and sediment cover characteristics in the steep and shallow sections of the five study channels.**

Discontinuity intensity and Schmidt Hammer values change with slope in the more thinly bedded sandstone rock, but not
in carbonate rock (Figure 6). Because the units are horizontally to near horizontally bedded, steeper stream channels cutting
through thinly bedded sandstone rock have more exposed bedding planes than channels with lower slopes. They also have
lower Schmidt hammer values (Figure 6a). However, discontinuity intensity and rebound values are invariant with slope in the
thickly bedded carbonate rock.

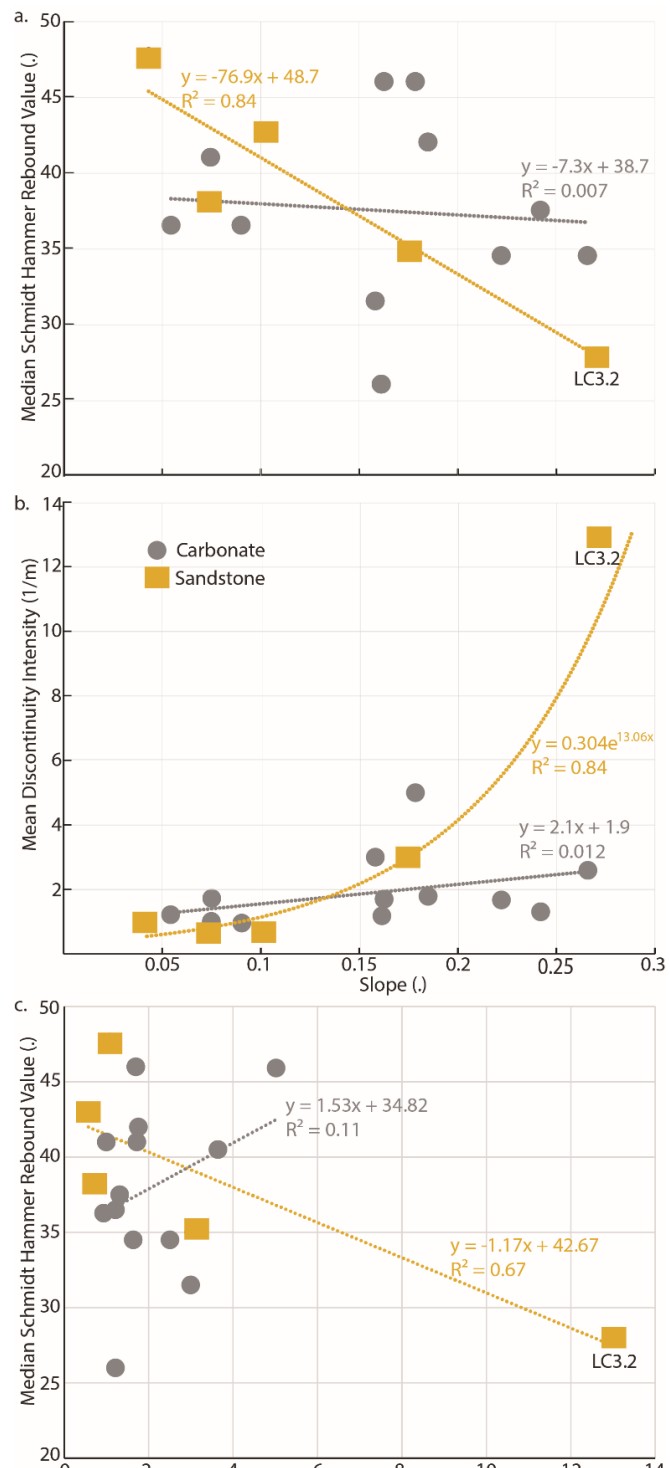

**Figure 6: a. Median Schmidt Hammer rebound value vs. channel slope b. Mean discontinuity intensity vs channel slope. We calculated slope over a distance of 75 m downstream and 75 m upstream of each reach. C. Median Schmidt Hammer values vs. Mean discontinuity intensity. All plots show data for 5 sandstone and 11 carbonate reaches. LC3.2, which was highlighted in Figure 2 and shown in Figure 4, is labelled.**

The average discontinuity intensity and Schmidt Hammer values from the thinly bedded sandstone in the steep channel section, where more bedding planes are exposed than in carbonate reaches, is 7.98 m$^{-1}$ (n = 2 reaches, standard deviation σ = 5.04) and 31.6 (n = 61, σ = 9.5) respectively. The average discontinuity intensity of the thickly bedded carbonate in the steep channel section is 2.34 m$^{-1}$ (n = 6, σ = 0.56), and they have an average Schmidt Hammer value of 36.1 (n = 240, σ = 10.8). Within the upstream channel sections, the reaches have a shallower slope with fewer exposed bedding planes per channel distance. In the shallower sandstone reaches, measured discontinuity intensity is smaller, 0.77 m$^{-1}$ (n = 3, σ = 0.16), but average Schmidt Hammer values are larger, 41.7 (n = 88, σ = 9.1), in comparison with the sandstone in the steeper section. Carbonate reaches in the shallow channel sections have a slightly higher discontinuity intensity of 1.51 m$^{-1}$ (n = 6, σ = 0.32) and average Schmidt Hammer value of 37.1 (n = 90, σ = 9.3) in comparison with the shallow sandstone reaches. In carbonates, discontinuity intensity and Schmidt Hammer values are essentially uncorrelated with channel slope.

Mean Discontinuity Intensity Values (1/m)

a.

| | Lithology | | |
| --- | --- | --- | --- |
| | Sandstone | Dolomite | Delta |
| Shallow | 0.77 | 1.22 | 0.45 |
| Steep | 7.98 | 2.28 | 5.70 |
| Delta | 7.22 | 1.06 | |

Mean Schmidt Hammer Values

b.

| | Lithology | | |
| --- | --- | --- | --- |
| | Sandstone | Dolomite | Delta |
| Shallow | 41.7 | 37.1 | **4.6** |
| Steep | 31.6 | 36.1 | **4.5** |
| Delta | **10.2** | *1.0* | |

Number of Rebound Values

c.

| | Lithology | |
| --- | --- | --- |
| | Sandstone | Dolomite |
| Shallow | 88 | 90 |
| Steep | 61 | 240 |

**Table 2: Table lists the a. discontinuity intensity values, b. mean Schmidt hammer values, and c. number of Schmidt hammer rebound values for sandstones and carbonates in the steep and shallow channel sections. Tables a. and b. include the differences (Delta) between the means of the same rock types or the same channel steepness. In table b., italicized delta values denote that the Schmidt hammer populations are statistically the same, bold delta values indicate that the populations are statistically different.**

We calculated four separate t-tests on Schmidt hammer measurements from the different rock types and channel sections
in Last Chance Canyon to determine if they are sampled from different populations. The null hypothesis is that the populations
of Schmidt hammer values in the carbonate and sandstone rocks are the same and/or do not vary between the lower steepness
(upstream) and higher steepness (downstream) reaches. This would indicate that the rock strength of the two different rock
types is statistically the same and support the idea that the erodibility does not vary between rock types or within rock types or
with channel steepness. Conversely, if the sampled Schmidt hammer values from different rock types are from statistically
different populations, this supports that the different rock types have different strengths and possibly different erodibilities.
We compared Schmidt hammer values between carbonate and sandstone reaches in the high ($t = 3.0$, $t\ critical = 2.6$, $\alpha =$
0.05) and low ($t = -3.4$, $t\ critical = 2.6$, $\alpha = 0.05$) $k_{sn}$ parts of the channel and found them both to be of different populations.
In other words, in the high $k_{sn}$ reaches of the channel, the sampled Schmidt hammer values from the carbonate and sandstone
rocks are from statistically different populations. The same is true in the low $k_{sn}$ reaches of the channel. The Schmidt hammer
values for sandstone reaches in the steep section were found to be statistically different from the Schmidt hammer values from
the sandstone in the shallow section ($t = -6.6$, $t\ critical = 2.6$, $\alpha = 0.05$). Schmidt hammer values for carbonate reaches in steep
and shallow sections were found to be from the same statistical population ($t = -1.1$, $t\ critical = 2.6$, $\alpha = 0.05$), which was the
null hypothesis. This was the only test of the four in which the null hypothesis was accepted and further demonstrates the lack
of strong correlation between channel slope and rock strength in carbonate reaches.

**4.3 Boulder Analysis**

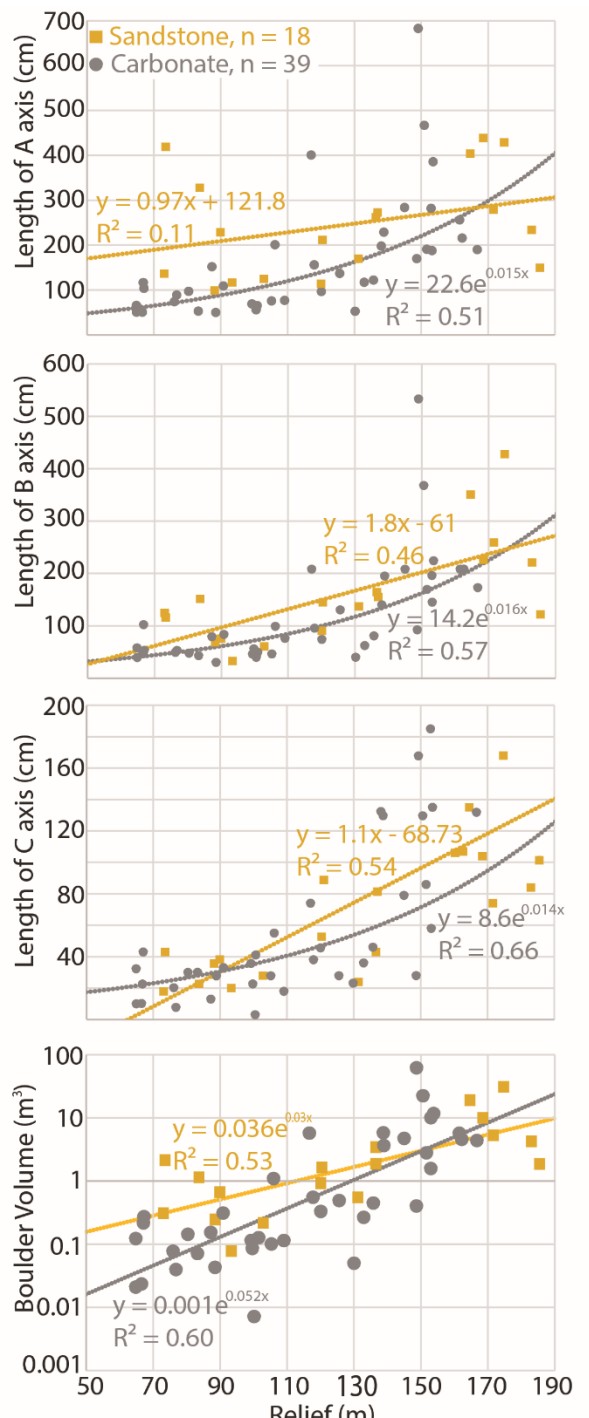

**Figure 7: Relief (calculated using a 500 m window) vs. the lengths of the a, b, and c axis, and boulder volume, calculated by multiplying the a, b, and c axis, for all boulders we measured in the field.**

As relief (calculated using a 500 m window) increases, the volume of the largest boulder in each reach tends to increase exponentially (Figure 7). Carbonate boulders tend to show a larger change in volume with relief than do sandstone boulders. Of the boulders we measured, 70% of the boulders in the high $k_{sn}$ section and 64% of the boulders in the low $k_{sn}$ channel section are carbonate. Boulder shape is also somewhat different between sandstones and carbonates. We used a simple shape factor c/a (i.e., the minimum boulder axis length divided by the maximum axis length) to quantify differences. Carbonate boulders had an average shape factor of 0.36 (n = 39, σ = 0.17), compared to sandstone boulders with an average shape factor of 0.29 (n = 19, σ = 0.18). Although the difference is small, carbonate boulders were on average more equidimensional (short and long axes more similar) while sandstone boulders were more elongate (a greater proportional difference between axes).

The correlation between the a, b, and c axes and relief is similar for the carbonate boulders we measured ($R^2 > 0.5$, and similar regression exponents from 0.014 to 0.016) (Figure 7). Lower relief corresponds to the upstream reaches. In the sandstone boulders we measured, the c axis correlates best with relief ($R^2 = 0.54$, regression slope of 1.1). The length of the b axis shows a slightly weaker relationship with relief ($R^2 = 0.46$, regression slope = 1.8) than the c axis. The length of the a axis ($R^2 = 0.11$, regression slope = 0.97) correlates poorly with relief. We fit an exponential trendline to the carbonate because it empirically gives a higher $R^2$ than a linear regression. Conversely, we fit a linear trendline to the sandstone boulders it gave a higher $R^2$ for the c axis. There was minimal difference between the $R^2$ values for exponential and linear fits for the a and b axis of sandstone boulders.

## 5 Discussion

Bedrock properties vary between lithologies and etch their signal on landscape morphology (Jansen et al., 2010; Scharf et al., 2013; Bursztyn et al., 2015; Yanites et al., 2017). In Last Chance canyon, differences in measured rock properties vary with changes in channel slope and local relief. Here, we introduce three key interpretations from our study. (1) Discontinuity intensity affects rock strength. We interpret that thickly bedded carbonate bedrock in our study area has high rock strength and low rock erodibility. In contrast, we interpret that the more thinly bedded sandstone rock (in comparison with the carbonate rock) has low rock strength and high rock erodibility. (2) We interpret that sediment input from hillslopes, and not rock properties on the channel bed, can set the rock erodibility when channels are armoured with sediment (following previous studies such as Duval et al., 2004; Johnson et al., 2009; Finnegan et al., 2017, Keen-Zebert et al., 2017). (3) We interpret that steep slopes can be sustained even where the channel bed is relatively weak sandstone because larger and more competent carbonate sediment armours the bed.

Putting these three interpretations together, we hypothesize that despite the change from low steepness upstream to high steepness downstream in our study channels, this is a relatively stable morphology in the current situation. We hypothesize that the channel sections with high steepness are not eroding due to the more massive carbonate units and the large, immobile

boulders armouring the channel, both of which lead to low channel erodibility. If the channel sections with high steepness are
not actively eroding, this creates a pinned base level for the low steepness channel sections upstream. This pinned base level
leads us to hypothesize that the high erodibility, low steepness upstream channels are also not eroding, creating an overall
stable morphology.

### 5.1 Lithology, Discontinuity Intensity, and Bed Slope

Local slope, bedding plane spacing, and fracture density control discontinuity intensity at the reach scale in Last Chance
canyon. If we assume that all bedding planes and fractures are horizontal, then for a given length of channel reach, steeper
reaches cut across more discontinuities than shallower reaches (Figure 8). We find that thinly bedded sandstone bedrock at our
field site has anisotropic properties. Layers are weaker (as measured by lower Schmidt hammer rebound values and higher
discontinuity intensities) when exposed in steep channels and are stronger in reaches with lower slopes that are more parallel
to bedding plane orientation (Weissel and Seidl, 1997) (Figure 6). When sandstone bedrock is eroded down to lower slopes
that are sub-parallel to bedding, then rock strength effectively increases and erodibility decreases, slowing further erosion.
This apparent reduction in discontinuity density holds true regardless of the vertical discontinuity spacing (Figure 8).
However, the apparent reduction in discontinuity intensity has less of an impact on the strength of the carbonate rock, because
even in the steep channel reaches the discontinuity intensity is low. We think this results in the carbonate rock strength being
independent of channel slope at our field site (Figure 6). Our statistical analysis of Schmidt hammer values from carbonate
bedrock in the shallow upstream and steep downstream channel sections confirmed that they are of the same population.

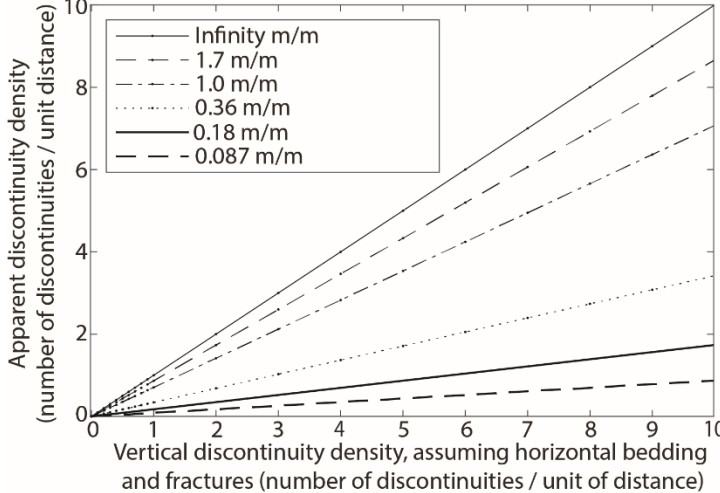

**Figure 8 – Relationship between measured discontinuity density along the bed (y axis) vs the discontinuity density if measured**
**on a face perpendicular to the discontinuities (x axis). Different lines represent channels with different slopes. Here the**
**discontinuities are modelled as perfectly horizontal, so a perpendicular face is vertical, or 90 degrees, or infinity m/m.**

There is a lack of exposed sandstone rock in channel reaches with higher slope. We only identified one sandstone reach in a steep downstream channel section. In surveyed channel reaches within the steeper downstream channel sections, we observed 0 to 7.8% of the channel to be exposed sandstone, and 74 to 100% alluvial cover (Figure 9; Table 1). In all five surveyed channels, the steeper downstream channel sections had more carbonate rock exposed than sandstone bedrock. We believe that our limited observation of sandstone in the steep channel reaches is because in comparison to the relatively hard carbonate rock, the relatively weak sandstone rock cannot maintain steep slopes. Where there is siliciclastic bedrock in the steep reaches, we interpret that it is armoured by boulders.

In summary, the landscape seemingly reflects the tendency of sandstone rock to erode to low slopes, creating a bi-modal landscape. In the shallow upstream channel section, there are more thinly bedded siliciclastic units exposed. In contrast, the steep channel section is mostly made up of thickly bedded carbonate rock or is inundated with sediment, resulting in a lower erodibility channel.

## 5.2 Lithology and Coarse Sediment Production

More thickly bedded and higher relief hillslopes contribute larger-sized and more geomorphically relevant boulders from the hillslopes to the channel (Neely et al., 2020) (Figure 7). The steep channel sections of Last Chance Canyon are incised into relatively narrow canyons, in comparison with the upstream, low steepness portions of the landscape. Hillslope derived sediment from the thickly bedded units in the canyon wall armors the channel bed in the steep reaches. We think these boulder deposits allow the relatively weak sandstone channel reaches to steepen through boulder deposition, as has been shown elsewhere (Shobe et al, 2016; Thaler and Covington, 2016; Chilton and Spotila, 2020). We assume that there are carbonate reaches that are also amorered in sediment. However, where bedrock is exposed in the steep channels, it is predominantly carbonate rocks, which are harder and presumably less erodible than the sandstone reaches (see subsection above). Within these steep channel sections which are inundated with sediment, we interpret that channel slope is somewhat independent of bedrock properties and instead depends on the amount, size, and competency of sediment armor sourced from proximal hillslopes. In other words, we think that the larger sediment armoring the steep reaches effectively decreases the erodibility of these reaches.

Bed thickness and fracture patterns control the initial size of sediment supplied by hillslopes to channels (Sklar et al., 2017; Verdian et al., 2020; Shobe et al., 2021). In Last Chance canyon, the maximum length of one axis of a boulder entering a channel from proximal hillslopes is controlled by the distance between bedding planes and fractures. In carbonate bedrock the distance between bedding planes tends to be longer than in sandstone bedrock. Where hillslope relief increases, bedrock units are thicker, and the length of the a, b, and c axes increases for the carbonate boulders (Figure 7). (We do not have measurements of discontinuity intensity from the hillslopes. Our observations were that steep hillslopes were primarily composed of massive carbonate.) In sandstone boulders, the c axis correlates with hillslope relief, the b axis length also correlates with relief, but to a lesser extent, and the a axis length does not demonstrate any relationship with relief. Because

sandstone bedrock is more thinly bedded, the c axis (shortest) will tend to reflect the distance between bedding planes from
the source rock.

The carbonate boulders are more equidimensional and have a higher average shape factor of 0.36 in comparison with the
sandstone boulders which have an average shape factor of 0.29. Although small, this difference in shape factor may reflect
how the distance between bedding planes affects sediment shape. Because a sediment grain tends to break across its shortest
axis, the more elongate sandstone boulders are less competent than carbonate boulders (Allan, 1997). Abrasion also reduces
boulder size and may decrease the size of elongate boulders more rapidly (e.g., Miller et al., 2014).  Also, this could be why
there were less sandstone than carbonate boulders. Of the 58 boulders we measured, 70% in the steep channel section and 64%
in the shallow were carbonate. Because carbonate bedrock is thickly bedded, boulders sourced from this bedrock tend to be
larger. Further, because the carbonate boulders are more equidimensional, they likely stay larger for longer than sandstone
boulders.

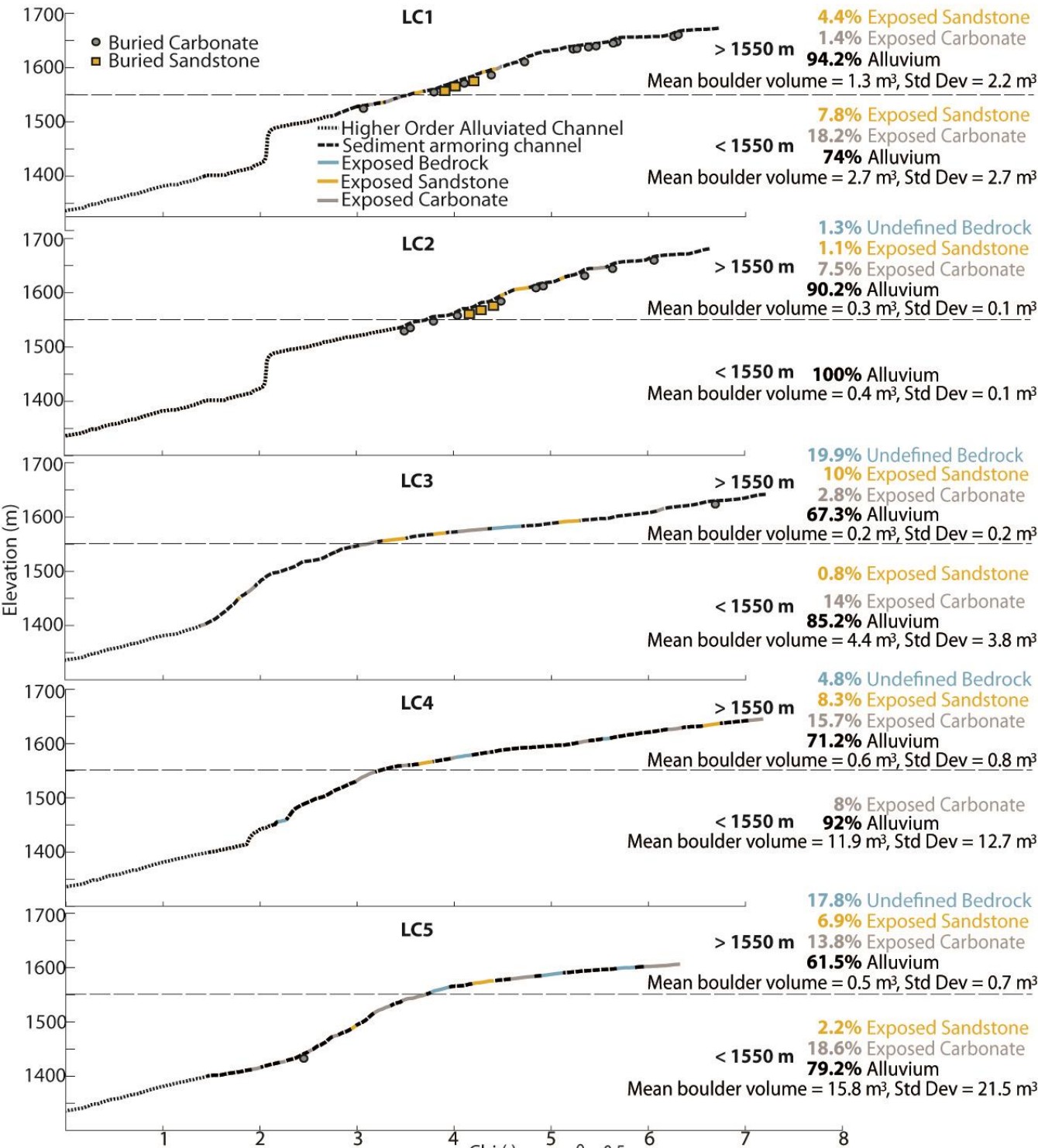

**Figure 9: Chi plots of LC1 - LC5 with exposed bedrock or sediment armored sections mapped. Where known, rock type beneath the sediment is shown by either a grey dot to indicate carbonate or a tan square to indicate sandstone. To the left of each channel, relevant statistics for each channel are displayed from 1400 - 1550m and above 1550 m. Average boulder volumes, which we measured in the field, above and below 1550 m elevation are shown along with corresponding standard deviations. High order alluviated channels are locations outside of our study area.**

### 5.3 Are Last Chance Canyon Channels Adjusted to Reflect Rock Properties?

We interpret that erosion in the steep reaches of our study channels is inhibited due to the presence of thick and resistant bedrock and large boulders that we interpret to be immobile. The downstream portions of our study channels are both steeper and have higher steepness indices than the upstream channel lengths (Figures 5, 9) and high steepness indices are thought to correlate with high erosion rates and/or less erodible rocks (Hilley and Arrowsmith, 2008). Although we do not have measurements of erosion rate in Last Chance canyon, we make the link between channel steepness and erodibility by assuming all channel reaches have a similar, low, erosion rate. In other parts of the Guadalupe Mountains, west of Last Chance canyon, erosion rates do not vary systematically with rock type, nor with slope (Tranel, 2020). We suggest that spatial variations in erodibility, rather than spatial variations in erosion rates, controls channel steepness in our study channels.

We further hypothesize that the upstream channel sections also have low erosion rates but for a different reason. These channel reaches have lower slope and lower channel steepness indices (Figures 5, 9). The upstream channel reaches are less armoured and have more sandstone exposed in the channel than their downstream reaches. These observations suggest that these upstream reaches are likely more erodible. Past erosion has reduced channel slopes leading to lower channel steepness.

The distinct upstream, low steepness channel and downstream high steepness channel is not consistent in all of our study channels. χ plots for channels LC 3, 4, and 5, demonstrate two well defined channel sections, where in the higher elevation, lower relief, and lower slope section above 1550 m there is more exposed bedrock, more exposed sandstone, less alluvium, and smaller boulders armoring the channel (Figure 9). In contrast, LC 1 and 2 lack the obvious transition from downstream steep section to upstream shallow section observed in LC 3, 4, and 5. We interpret that the less notable change in upstream steepness in LC 1 and 2 is due to the armoring of sandstone rock units and relative abundance (in comparison with LC 3, 4, and 5) alluvium above 1550 m in elevation. Lithology measurements from proximal hillslopes in LC 1 and 2 indicate that just above elevation 1550 m there are sandstone units in the channel, as there are in LC 3, 4, and 5, but they are buried by alluvium in LC 1 and 2 (Figure 9, Table 1). We note that the transition to a lower steepness occurs at a higher elevation in LC 1 and 2, at about 1640 m (Figure 5) and it may be less distinct in comparison with LC 3, 4, and 5. We do not know why there is more extensive armouring in LC 1 and 2 in comparison with LC 3, 4, and 5. One possibility for this armour is the outcropping of the Queen formation on the hillslopes above LC 1 and 2 but not above LC 3, 4, and 5 (Figure 2). Regardless of the reason, the fact that LC 1 and 2 remain steep even when the channel bed is sandstone supports our idea that sediment cover can hide the properties of the local bedrock and impact channel morphology

Through landscape evolution modelling using the stream power model (Equation 1), Forte et al. (2016) showed that where more erodible rocks upstream are underlain by less erodible rocks downstream, the upstream reaches can have an effectively pinned base level, such that channel steepnesses evolve to reflect the contrast in rock properties. Our overall interpretation of

the Last Chance Canyon landscape is consistent with bedrock properties exerting this type of control. We also note that Perne
et al. (2017) demonstrated that if topography is adjusted to bedrock erodibility in horizontally layered rocks, erosion rates
should only be consistent if measured parallel to the layering. We interpret the Last Chance Canyon landform to approximate
a steady state geometry, but relative to the horizontal bedding over time (Perne and Covington, 2017). Our bedrock properties
data also illustrate challenges in directly linking measurable rock properties to bedrock channel reach erodibility. However,
our data also suggest that coarse sediment—rarely mobile boulders which reflect nearby bedrock eroding from hillslopes, but
not the local channel bed itself—are a key mechanism by which lithologic contrasts are expressed in this landscape. Future
work could explore how boulder transport may move and disperse zones of lithologic control downstream from boulder source
areas. Regardless, we interpret that the bimodal topography in Last Chance Canyon– low to high steepness channels and less
steep to steeper hillslopes - has evolved to reflect the rock properties of the two dominant lithologies, both locally and non-
locally.
**5.4 The Guadalupe Mountains Beyond Last Chance Canyon**
Our ability to hypothesize about the impact of rock properties on landscape morphology in Last Chance Canyon required
extensive observations and field and lab measurements. Even in our small study area of 8 km$^2$, the morphology of channels
LC 1 and 2 varies from LC 3, 4, and 5 above 1550 m. Our measurements of sediment cover and buried rock type allowed us
to hypothesize why these channels are different, despite incising into the same stratigraphic units. This led to a consistent
process interpretation, despite different landscape morphologies.
South of Last Chance Canyon, in the main escarpment of the Guadalupe mountains where channels drain to the southeast
(Figure 1), the reef complex led to more massive carbonate deposits. Those deposits now form prominent peaks, such as El
Capitan, in the southern-most part of the Guadalupe mountains. The longevity of these peaks and the strength of the deposits
that form them suggests that the reef complex deposits are less erodible than surrounding deposits. Given the complex local
and non-local role of rock properties on channel morphology and the different rock units that outcrop beyond Last Chance
Canyon, we are hesitant to project our interpretations of how rock properties impact channel morphology to the greater
Guadalupe Mountains. However, we think that the methods laid out in this paper, along with the modeling frameworks of how
rock erodibility contrasts impact channel evolution (Forte et al., 2016; Perne et al., 2017), present a guide for deconvolving
the complex role of rock properties on channel morphology in the broader Guadalupe Mountains and beyond.
**6 Conclusions**
We present several observations about the effects of rock properties on bedrock channel steepness in tributaries of Last
Chance canyon. We suggest that discontinuity intensity influences channel steepness. Streams steepen across carbonate units
that have thicker beds and lower discontinuity intensities in comparison with the sandstone in this area. Conversely, channel
steepness is lower in channel reaches incised into thinly bedded sandstone units with higher discontinuity intensity.

The extent of sediment cover and the size of boulders in the channel also impacts channel morphology. More thickly bedded carbonate bedrock on the hillslopes contributes larger alluvium to the channel. This coarse carbonate sediment armours both the more and less thickly bedded bedrock and smooths channel slope across reaches with different lithologies and discontinuity intensities. In Last Chance canyon, channel sections that contain larger carbonate alluvium are generally steeper even if the channel bed is siliciclastic with high discontinuity intensity.

Finally, we interpret that the study reaches have evolved to a relatively stable morphology adjusted to bedrock erodibility and local coarse sediment supply. The more erodible shallow channel reaches at the top of Last Chance canyon have a base level that is pinned by the steep, and less erodible, channel downstream. Any downcutting of the steep channel reaches downstream will likely result in corresponding lowering in the lower slope and more erodible reaches upstream, maintaining a similar channel profile through time.

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
