# Peer review of "Building a Bimodal Landscape with Varying Bed Thicknesses in Last"

_EGUsphere, 2022_

## Author Response (AR1)

We appreciate the effort, time, and thoughtful and constructive ideas presented by the reviewers and the AE. We have implemented most of the suggested changes and agree that they have improved the clarity and presentation of the manuscript.

Following the suggestions from both reviewers we have reorganized and edited the introduction and discussion sections. We have added more detail to the methods section. We have made changes to the figures and added figures and tables. In summary, the entire paper has been rewritten. In fact, we have two distinct edited versions of the document with track changes because it was getting impossible to read and edit the first version.

Thank you for your time. We hope you will find this a better manuscript.

Below are our line-by-line responses. Please note that we have made a good faith attempt to address all the reviewer comments. In some cases, after so many edits and moving of text by three different authors, we couldn't find the location of the reviewer's suggested edits even with the line numbers, so we assumed the change was made or the text was removed.

**Reviewer1:**

1. general comments
* * *
Overall, this manuscript deals with and disusses a relavant and well-set study topic fitting the current state of research on landscape development based on lithological setting. There is a good and clear hypothesis, though, there are several issues to be faced in the current version of the paper:

- the introduction needs a distinct backing-up and reasoning by more literature; the discussion needs to be straigthlined

**We have added more citations to the introduction and reorganized the discussion.**

- there is need of more discussion why the one outlier (L3.2) is valid as it is basic for some results/interpretations

- the outcomes developed in the discussion need to be more streamlined and several controdictions need to be cleared

**Discussion was rewritten**

- several figure panels should be combined to ease the interpretations

**We combined two map figures into one.**

Thus, this study findings would generally be worth publishing after adressing the above mentioned tasks - below there are several detailed hints/notes/suggestions on how to address them - both scientificaly and technically.

**2. specific comments**
* * *
L1 use a more expressive statement

We are unsure what this means. Assuming the reviewer is asking for a title change, we modified the title to one which we think is more expressive, from "Building a Bimodal Landscape with Varying Bed Thicknesses in Last Chance Canyon, New Mexico" to "Building a Bimodal Landscape: Bedrock Lithology and Bed Thickness Controls on Channel Morphology in Last Chance Canyon, New Mexico, USA"

L23 abstract: we believe?

**Changed to "We interpret".**

INTRO

L27 (only) little debate?

**Edited to remove the "little debate" statement**

L35 also Shobe++2021, GSA Bulletin

**Added citation**

L37 needs definition what the ksn actually is (physically) or general description of channel profile descriptors (as they are more defined in the methods)

**We now do not mention ksn in the introduction, and instead define it in the Methods section.**

L45 needs info on the geochemical methods and data as background, there is no info yet

**In editing the introduction we cut the mention of geochemistry.**

L47-L49 why inverse (physical explanation)?

In editing the introduction we cut this description.

L54 also Scott&Wohl2019, ESPL

**Added citation**

L56 cf. Bursztyn++2015, EPSL

**Added citation**

L57 to fluvial geomorphologists, too!

**Introduction was overhauled**

L58 intro of sediment availability, sediment size, btools and cover, and discharge variability is missing, also channel width vs. steepness is not mentioned - these topics are fundamental in this context!

We now acknowledge the importance of all of these topics in the introduction. We note that our data and analyses do not explore or provide insights into either discharge variability or width-steepness relations, although we recognize the importance of these in eroding bedrock channels.

L59 reason, why foirst-order channels (aslo, what are these)

We added a reference to the definition of first and second-order channels in the methods section.

L61 too colloquial text - e.g., "find rock mineralogy" ...

**Introduction was overhauled.**

L65 landscape or river channels?

**Introduction was overhauled.**

L68ff higher elevation in this scanario!

**Introduction was overhauled.**

L70 is this for a ~steady state case the weaker erodibility may be deducted even? (cf. Mitchel&Yanites2021, ESURF)

**Introduction was overhauled.**

**FIELD**

L71 climate (so Kc) is assumably constant, i.e. can be ignored for this analysis? Added

L85 how about the sediment (size distribution, lithological partition) in the investigated reaches?

**We are unsure what this means.**

METHODS

L105 Xi needs references

added

L109f which DEM; why 75m?

**description added**

L112 are the San Gabriel Mountains reasonably comparable to your site (concerning chanel geometry, lithology, grainsizes, climate etc.)?

**This description was changed to no longer mention the San Gabriel Mountains.**

L117 a metric interval would be more tangible for the community

**changed**

L118 why (only) the largest boulder - is this significant of anything (e.g., cover)? What is the relation to / meaning for smaller grainsizes?

**explained in text**

L121 which unit

**We were unsure what this meant, but we explained the units on Schmidt Hammer in the text.**

L133 define what plucking is, and why it is important here

done

L142 posting? you mean resolution?

**changed**

fig4b end of caption is unclear; line colors in c and d are hard to differentiate - take a color-blind friendly range; indicate the Sitting Bull Falls in 4a (is this at L3.2?); also having notes on which channel holds which lithology (refer to fig.2) would be very helpful to get the point

**All of the changes to figure 4 were made, except we did not add anything about lithology to the figure that was formally 4, now figure 5.**

fig5 how does a plot of discontinuity vs. Schmidt Hammer Rebound look like? What do the results tell you?

**Figure added.**

**RESULTS**

L187f the carbonate values are not much dfferent between steep and shallower sections

**Yes. We added a sentence that says this explicitly**

L201 how does the pattern look like if you clipp the actual in-channel boulders from the 500m window

**We are not sure what you mean by this. In editing, we also cut the sentence that was previously line 201.**

L208ff I assume you refer to fig.7 - you state there "all boulders", but these are 'only' the largest boulders per reach, right? So, at least your result is not generally valid?

We now directly acknowledge that this for the "boulders we measured". In the methods section we also say that this method may introduce bias in terms of the measured vs. actual size distribution.

**DISCUSSION**

L221ff for the 5 points refer back to the figures, respectively!

We respectfully disagree; we refer to figures elsewhere in the discussion (and throughout the paper of course), but the key points do not seem to us to distinctly refer to different separate figures.

L226 Shobe++2016, GRL

**Added**

L229 didn't you measure larger Schmidt hammer values in the shallower sections above attesting them harder rock?

**This is true for the sandstones but not for the carbonates.**

L231 that may be valid for you lithologies, but not generally

**Agreed, reworded to emphasize our particular field site.**

L236 I don't get this reasoning ...

**We cut this sentence, and agree that the argument was not strong.**

L238 you mean there only is one data point for steep slopes that determines your whole interpretation above - correct; you say here you ignore it - so what about all the results?; why is this outlier there (is it

an transient knickpoint? this would contradict L227ff)

**L244ff several repetitions, reduce**

**In rewriting the discussion, including this part of it, we have tried to remove repetitions while still explaining points clearly.**

L252 so why are there steep vs. shallow carbonate sections

In editing we have removed this line. The explanation would be that rocks are interbedded and so there are still carbonate layers in sections of the stratigraphy that are overall weaker.

L261 so erosion is focusing on the steep sections (until they are shallow enough to hold cover?) - both on carbonate and sandstone? Though, you say the opposite in L296ff

We are not sure what this means.

L267 not by fracture distance?

**Fracture distance is generally controlled by bedding thickness**

L272 so then - how is the correlation between bedding thickness with local rock dimensions

**We do not have measuremets of just bedding thickness, only discontinuity intensity. But in general discontinuity intensity is partly controlled by bedding thickness.**

L272ff several repetitions, shorten; though you could repeat the bedding thickness values/orders for better evaluation of your discussions!

**We have tried to remove repetitions**

L283ff this section is missplaced and also repeats a lot; have this earlier in the interpretation - also fig9 partly repeats fig.4cd and should not show up here in the discussion; coul go to the supplement (or maybe show one example of the rock exposure as a panel in fig.4)

We respectfully disagree, and have left this figure in the discussion because we feel it is important and useful as a summary figure that lets us talk about data from the rest of the paper. It also emphasizes the difference in bedrock exposure vs. alluviated channels. We also note that Reviewer 2 likes this figure, and said "Figure 9: Lots of important observations here that I did not appreciate my first couple times through the manuscript." For this reason we feel like it is important to keep it in the main paper, and to show all of the profiles.

L296 contradicts L304f (and L261) - confusing and circular these two last paragraphs; solve for a reasonable, streamlined and consitent interpretation at one place in the text

**We edited this part of the text significantly, including removing line 304, in order to streamline as suggested and to reduce the appearance of circular arguments.**

**CONLCUSIONS**

L318 need to mention Carbonates here?

Yes, we replaced sedimentary with carbonate units, but also emphasize that the carbonates have "thicker beds and lower discontinuity intensities in comparison with the sandstone in this area"

3. technical corrections

L39 not necessary

changed

L86/88 repetition

fixed

fig1/2 combine into panels

done

L146 why 40 foot and not [m]?

explained

L189 for on

changed

L209 combine fig.6 and the panels of fig.8 into 4 panels; fig.7 is wrong-placed **combined and moved**

fig8 the caption indicates fig6 is added as a panel - do that

see above

fig9 caption: left is right ...; what are the dots?; what are "high-order alluviated channels"; rock-coloring is hard to differentiate

**We changed the caption to better describe the dots and the higher-order alluviated channels.**

Citation: https://doi.org/10.5194/egusphere-2022-1285-RC1

**Reviewer2:**
* * *
**General comments:**

Building a Bimodal Landscape with Varying Bed Thicknesses in Last Chance Canyon, New Mexico by Anderson et al. presents a detailed analysis of 5 catchments in the Guadalupe Mountains in southern New Mexico, USA. The authors used digital elevation model analysis, field topographic surveys, and field derived estimates of bedrock properties to interrogate how lithology influences bedrock river morphology via its dual roles on setting coarse sediment delivery to channels and the bedrock erodibility coefficient. In this setting, which is dominated by horizontally stratified carbonates and sandstones, the authors find that: 1. Thick carbonates are less erodible than sandstones due to fewer discontinuities and 2. simple interpretations of bedrock erodibility in the channels are confounded by the delivery of coarse, carbonate sediment that armors more erodible sandstone reaches downstream. These two observations lead the authors to conclude that steeper, armored reaches downstream have evolved towards a relatively stable morphology such that slowly eroding upstream reaches are experiencing a constant base level. The fine resolution mapping of rock strength, topography, and coarse sediment are both hard to obtain and immensely valuable to testing our understanding for how rivers incise (or not) into bedrock. As such, this analysis is very much suited to Earth Surface Dynamics community and will provide a useful empirical dataset to the geomorphology literature as a whole. That said, there are two modest sets of revisions I think will help improve the impact of this analysis and manuscript:

- 1. A broader context for how field sites fit into the landscape.
- 2. Some tightening of the text to clarify major claims and implications.

I do not think there is a need for any major new analysis and thus believe this manuscript will be ready for publication pending minor to moderate revisions. Below, I briefly expand on these two main points and then provide a list of specific line-by-line comments that may help during revisions.

**Broader Context**

There is always a trade-off between resolution and coverage, whereby this study is an important contribution on the resolution side. The hard-won field data reported in this study merits publication alone by providing very detailed observations for how first order channels erode into sedimentary rock with large erodibility contrasts. That said, I found myself wanting to know more about why they picked the channels they did and how representative the patterns they observed are to the Guadalupe Mountains as whole. Are any of the reach-scale patterns in rock erodibility and coarse sediment

production encoded in the geologic units such that it makes predictions for other first order stream in the landscape? Given the relatively small scale of the watersheds analyzed (based on Google Earth they seem to be of order 20 km2; please add table with watershed characteristics), I think it will be useful to see how patterns in channel steepness and lithology translate to rivers more broadly in this landscape. Though it might be beyond the scope to analyze regional data, I think it is still important to show this broader context in mapview in the Introduction.

We have spent time hiking and observing and many of the channels across the Guadalupe mountains. The short answer to the question of whether the patterns we observe are also present elsewhere is yes and no. There are locations where small streams are deeply incised and lined with boulders that we think are sourced from the more massive sedimentary rocks on the surrounding hillslopes. However, there are also locations with a different baselevel in which the lower-order valleys have less relief and less cover. We have also observed that the same lithologic units have spatially variable rock properties. Further subtle changes in climate make a large different in local vegetation and weathering. Thus, there are numerous factors controlling rock properties and channel morphology.

The scope of this manuscript was to identify variables that control bedrock morphology in this setting and the best methods to quantify these variables. There were plenty of measurements made that were not helpful for understanding landscape morphology. Next steps from this study could be to make similar methods in different watersheds across the range, but this is beyond the scope of this study.

Ultimately we focused on the study area that we did because it was (relatively) easily accessible. In this region accessibility considerations include quality of roads leading to the field area, length and difficulty of hike from road to study channels, and whether the area was public and could be sampled. We did not have a permit to remove rocks from the national parks.

We have added more explanation of the context of our study channels within the broader landscape in the Study Area section of the manuscript. We also added two paragraphs at the end of the discussion about how to think more broadly about our results. We chose not to add a geologic map of the entire area because it is difficult to interpret the different units over a large area, and because of the spatial variability in rock properties, we do not think it is helpful. However, we have added a topographic map of the larger area as figure 1.

**Tighter narrative**

While the overall structure of the manuscript is strong and the writing is relatively clear, I think this manuscript could benefit from one more round of careful editing. In particular, the Introduction could use a bit of expansion and the Discussion could benefit from some re-structuring. For the Introduction, I think perhaps framing the problem more centrally around the work of Forte et al. (2016) and Thaler & Covington (2016) could be useful, as elements of this study reiterate findings from both prior studies. By addressing the quadruple challenges of horizontal rock units, complex rock strength assessment, strong erodibility contrasts, and complex interactions with coarse sediment supply, I think it is important to communicate how important the high-resolution data these authors are collecting is.

Thank you for these helpful suggestions. We have completely rewritten the introduction. Forte et al. (2016) now features prominently (3rd paragraph) as well as the related paper Perne and Covington

**(2017). Thaler and Covington (2016) is described in detail in the 5th paragraph. We agree that this reorganization and focus improves the communication.**

The Discussion currently contains lots of good insights, though I found it a bit wandering in places and redundant in others. As currently structured, L218-230 is an overview. L231-261 attempts to explain how the reach-scale patterns are linked to rock strength and coarse sediment cover. L265-279 explains differences in boulder production between rock types. L283-291 revisits the role of coarse sediment armoring for each of the surveyed channels. L296-315 builds on this to articulate why steepness may not be correlated with incision and why this whole system may be relatively static. Perhaps adding a couple of subheadings could aid in organizing this part of the narrative (e.g., 5.1 Lithology and coarse sediment production and 5.2 Implications for landscape stability).

**We have significantly reorganized the discussion and have added subheadings.**

**Line-item edits**

L15 and throughout: Consider replacing the acronym DEM with DSM. I do not mean to be too picky here, but as I understand it, the authors are generating digital surface models (DSMs) since they are not filtering vegetation. Flying around Google Earth makes me think this is a pretty minor source of uncertainty in either derived hillslope or river metrics. That said, it is worth being precise in the language around this so that the authors can make that point.

**Thank you. We have changed the use of DEM and SSM throughout. In some locations we are using the USGS DEM, in others our own data, which we now refer to as DSM.**

L15: Consider replacing 'drone photos' with 'drone and ground-based photogrammetry.' This might require tweaking some of the sentences following, but it seems to me the authors would want to highlight the GoPro data for mapping bedrock discontinuities.

**done**

L22: Consider replacing '...dampens...channel steepness.' with '...dampens steepness contrasts across rock types.'

**done**

L24: Delete 'essentially'

**Done.**

L44-45: '...we could estimate...' I didn't quite understand what the latter part of this sentence was getting at. Kc is undoubtedly an important piece of the puzzle but independently constraining it wouldn't be enough to predict incision from topography for many reasons, including some of the sediment dynamics ones the authors argue for here.

**Introduction was completely changed.**

L46: Akward phrasing. Not sure what is meant by empirical definition of eq. 1.

**Introduction was completely changed.**

L53: Consider replacing 'variables that must' with 'bedrock properties that should'.

**Introduction was completely changed.**

L61: Replace 'carbonite' with 'carbonate'

**done**

L62: Check use of commas here and throughout.

**Done, although there are differences among the authors about preferred comma usage.**

L68: Forte et al. (2016) is a good citation here, but I think also a good place to bring in Thaler & Convington (2016). I also think the authors could better elaborate on the ideas gleaned from these prior studies. Forte et al. (2016) had no sediment. Thaler & Covington (2016) argue for the fundamental importance of coarse sediment in armoring channels (funnily enough the hard rock at their site was sandstone and the weak rock was carbonate). These two prior studies are complementary though in that a key ingredient was horizontal rock units. The authors here have presented detailed field measurements for one scenario (i.e., hard over weak), and see elements of both models. In my view, this last paragraph of the introduction could use some re-framing around these ideas to help setup later interpretations of their data.

**Introduction was completely changed.**

L71: The field area is awesome, and I understand the focus on where data was collected. That said, this section could use a figure that shows the regional geomorphic context. Ideally, I would love to see this regional context carried through the manuscript by introducing the broader river network here and then relating geology to channel steepness below. I recognize that this may be beyond the scope of this study. As such, I suggest at least putting a regional map figure in this section showing the study area, geology, topography, and river network.

**We now include a regional topographic map.**

L89: Please capitalize 'Figure.' This is the first example I noticed, but it appears throughout the manuscript.

**done**

L89-90: Awkward sentence. Perhaps something like 'Rock unit descriptions from published geologic maps are not at the scale needed for us to constrain rock strength.'

**changed**

L97: Lat and long for LC3.2 is also in the Figure 3 caption so I don't think it is needed here. Also, it seems the authors have a naming convention for the GoPro5 imagery LCx.y where x is the channel number and y is the site along the channel. It might be useful to provide a summary table that explains this naming convention and points to where the datasets can be accessed.

We changed the wording in the caption so that it says LC3, rather than LC3.2. We don't refer to the reach subsections elsewhere, and it wasn't necessary here either, so we don't explain our naming convention. The channels were simply numbered based on location. We kept lat long in here because it is the lat long of this location.

L105: Suggest deleting 'like ksn'. Also, the authors may want to explain in a bit more depth what chi is for the uninitiated.

**done**

L111: I think this choice of 500 m (is this the radius or diameter?) to calculate hillslope relief is fine, though I was bit puzzled by the citation of DiBiase et al. (2010). That prior study argued that a 2.5-km radius tracked with channel steepness (and thus fluvial relief) and that

**We removed "relatively high", and added R^2 values to the main text in addition to being presented on the figure.**

209 I do not see where the judgement of conversely comes from. In the preceding sentence, you have not given any contradicting information.

**We edited the text to remove the Conversely.**

210 Thanks for the quantitative information! :o)

210 What is the parameter m? What kind of relationship did you fit?

**We reworded the text to remove the undefined variable m and say that are reporting exponents or slopes in the regressions.**

210 Capitalize 'the'. Replace 'demonstrates' with 'shows' or something similar.

**Changed**

212 better than what? Is a 'better fit' a good criterion for choosing one regression model or another?

We removed the subjective "better", and now just state that we chose the exponential fit because of its higher R^2 value. We now also acknowledge directly that the choice of equations to fit is empirical.

214 I neither understand what you mean by 'slightly more' nor by 'equidimensional' in this sentence. The quantitative information in the second half of the sentence is sufficient; consider deleting the first half.

We reworded the text describing boulder shape. We now start with the quantitative numbers of the shape factor. Nonetheless, we still describe what the numbers mean, but provide more explanation in terms of the shape factor we use. In particular, we say that equidimensional means that the short and long axes were more similar, while more elongate means that there was a greater proportional difference between the short and long axes.

219 Be specific, there are many more rock properties than you have measured!

Yes, for sure. We have stated which ones we measured many times already and go on to restate what properties we are talking about. This is just an opening sentence and we thought it was OK to start big and then be more descriptive. Did you want us to remove the references?

220 positive or negative correlation?

**changed to vary.**

220 In the list, make clear what is interpretation and what is the observation that the interpretation rests on. For example, under (1), the second half of the sentence refers to an observation.

**For this and the next 4 comments, we rewrote this paragraph to seperate our interpretions of the data from what we hypothesize.**

221 higher than what?

222 (2) is just the corollary of (1), the point does not give an interpretation.

223 (3) Please provide an argument to justify the interpretation.

228 Is this an interpretation (as announced in line 220) or a hypothesis?

240-252 Here, my impression is that there is a mixture of observations, interpretations, and hypotheses, which are not clearly labelled and / or separated, and some of the reasoning behind it is implicit. Please rewrite to clarify and make reasoning explicit.

**We have rewritten most of the discussion with this in mind.**

243 What is the difference in the carbonate and sandstone ways of anisotropy?

**We do not understand what the reviewer is asking.**

258-261 not sure whether I follow the reasoning here. This could be connected to current theoretical understanding of how the channels work (e.g., graded stream paradigm, channel morphology evolves to match erosion to uplift). The recent publication by Nativ et al. may be helpful to make this connection (doi: 10.1029/2021JF006537).

**This was all reworded.**

259 In my understanding, the causal relationship would be the other way round. I.e., the stream has a need for erosion, because of uplift or baselevel drop. It adjusts its morphological state – e.g., slope and cover – to match this need. Of course, this works only if the observed situation reflects a steady state. Yet, if it is not in a steady state, why would any observed relations be informative?

**I think this was completely reworded. I can't find a scrap of this sentence.**

260 What is a "potential for erosion"? How would you measure it? This seems to me to be a theoretically-laden concept disconnected from the underlying theory. Can you make the concept behind it explicit?

**This is gone.**

265 How do you know? Maybe the causality is the other way round.

**We are unsure what this means. Do you mean that sediment grain size in the bed impacts the bedding of the surrounding rocks?**

**275 Reference?**

**added.**

273 and following: are the differences significant? What does the word 'subtle' (line 274) mean in this context?

**changed.**

277 The relative fractions should be controlled by delivery, transportability and size reduction. The latter two determine average residence times. The argument presented here seems plausible, but also incomplete.

**We interpret the large boulders to be immobile.**

279 the underlying concept behind this reasoning seems to be that boulders diminish in size mainly by fracture. What about abrasion?

We now acknowledge abrasion for reducing boulder size and modifying shape, and cite Miller et al., 2014.

287 Interpretation or hypothesis? If the latter, how will you test it?

**Changed to interpret**

290-291 Can you make your observation, reasoning and interpretation explicit here?

**This was reworded and expanded.**

296 interpretation or hypothesis?

**Changed to interpretation ("We interpret...")**

310 The connection to the modelling papers is not clear and needs more detail. This connection has been highlighted in the final paragraph of the introduction, indicating that the authors find it particularly important, and the level of the discussion here should reflect and reason this importance.

As suggested by other reviewers, the introduction frames the relevance of collecting field data even more in terms of previous modeling studies. While our main focus remains on field data, we return to the modeling in the Discussion section. We do this to put our interpretations of bedrock and coarse sediment controls in the framework of those papers.

313 I think this statement is an important assumption driving both the way you present the data and their interpretation. Maybe you can move this to an earlier point in the article?

Following this suggestion, we added a short paragraph to the end of section 4.1 which states this, to hopefully guide the reader more clearly through our data and arguments.

Citation: https://doi.org/10.5194/egusphere-2022-1285-EC1